# Implications of Selective Autophagy Dysfunction for ALS Pathology

**DOI:** 10.3390/cells9020381

**Published:** 2020-02-07

**Authors:** Emiliano Vicencio, Sebastián Beltrán, Luis Labrador, Patricio Manque, Melissa Nassif, Ute Woehlbier

**Affiliations:** 1Center for Integrative Biology, Faculty of Science, Universidad Mayor, Camino la Piramide 5750, Huechuraba 8580745, Santiago, Chile; ev.fibonacci@gmail.com (E.V.); sebastian.beltran.v@gmail.com (S.B.); alexardy@gmail.com (L.L.); patricio.manque@umayor.cl (P.M.); 2Center for Genomics and Bioinformatics, Faculty of Science, Universidad Mayor, Camino la Piramide 5750, Huechuraba 8580745, Santiago, Chile; 3Escuela de Biotecnología, Facultad de Ciencias, Universidad Mayor, Camino la Piramide 5750, Huechuraba 8580745, Santiago, Chile

**Keywords:** autophagy, amyotrophic lateral sclerosis, p62, TBK1, ubiquilin2, optineurin, NBR1

## Abstract

Amyotrophic lateral sclerosis (ALS) is a lethal neurodegenerative disorder that progressively affects motor neurons in the brain and spinal cord. Due to the biological complexity of the disease, its etiology remains unknown. Several cellular mechanisms involved in the neurodegenerative process in ALS have been found, including the loss of RNA and protein homeostasis, as well as mitochondrial dysfunction. Insoluble protein aggregates, damaged mitochondria, and stress granules, which contain RNA and protein components, are recognized and degraded by the autophagy machinery in a process known as selective autophagy. Autophagy is a highly dynamic process whose dysregulation has now been associated with neurodegenerative diseases, including ALS, by numerous studies. In ALS, the autophagy process has been found deregulated in both familial and sporadic cases of the disease. Likewise, mutations in genes coding for proteins involved in the autophagy machinery have been reported in ALS patients, including selective autophagy receptors. In this review, we focus on the role of selective autophagy in ALS pathology.

## 1. Introduction

### 1.1. Amyotrophic Lateral Sclerosis (ALS)

Amyotrophic lateral sclerosis (ALS) is one of the most common adult diseases characterized by the loss of upper and lower motor neurons in the brain, brainstem, and spinal cord, causing progressive paralysis of the voluntary muscles [1]. Like other neurodegenerative diseases, ALS is a complex disease with a growing number of genes and loci associated with it [2,3,4,5]. Most cases of ALS are classified as sporadic (sALS), while 5–10% are familial (fALS), due to a genetic component segregating in the family [6]. The first mutations identified as a cause of fALS were mapped to SOD1 (superoxide dismutase 1), which account to date for about 2% of total cases [1]. Mutations in SOD1 trigger its misfolding and aggregation, a phenomenon also observed for its wild-type protein in sALS cases [7,8]. Mutations in the *TARDBP* gene, coding for Tar DNA binding protein 43 (TDP43), one of the most common components of protein aggregates in ALS cases, were also identified. In recent years, the number of genes associated with ALS has significantly increased (reviewed in [1]). Mutations in six genes can explain about 60–70% of fALS and about 10% of sALS cases: *SOD1*, *TARDBP*, *FUS* (fused in sarcoma), *VCP* (valosin-containing protein), *OPTN* (optineurin), and *C9orf72* [4,9]. Other less-frequent genes associated with ALS include *VAPB* (vesicle-associated membrane protein-associated protein B), *TBK1* (*TANK-binding kinase 1)*, *UBQLN2* (ubiquilin-2) and *SQSTM1* (sequestosome 1)*/p62*.

Mechanisms found impaired in ALS mainly involve four pathways: RNA metabolism and translation, protein quality control and degradation, mitochondrial function, and transport and trafficking via the cytoskeleton within and between compartments [4,9,10,11]. As an intracellular hallmark of ALS, motor neurons present an accumulation of protein aggregates and dysfunctional organelles, such as mitochondria. Cytosolic aggregation of TDP43 is present in the majority of ALS (97%) and about half of frontotemporal dementia (FTD) cases [12]. Other proteins that accumulate in ALS are mutant FUS, SOD1, and C9orf72. In the case of C9orf72, G4C2 intronic repeat expansions lead to the expression of repeat-associated non-ATG-mediated (RAN) peptides, known to form toxic intracellular aggregates [13]. 

Accumulated proteins have been considered a consequence of the impairment in protein quality control pathways, including translational and degradation processes. Intracellular accumulation of misfolded protein aggregates and damaged organelles are toxic to neurons. Furthermore, several of the genes in which mutations are associated with ALS encode proteins that take part in the protein quality control machinery, hence are part of the disease etiology themselves. 

### 1.2. Autophagy Pathways 

Autophagy is a highly conserved degradative mechanism that contributes to maintaining cellular homeostasis. Three kinds of autophagy pathways were described in eukaryotic cells: macroautophagy, chaperone-mediated autophagy (CMA), and microautophagy [14,15]. 

CMA cargos are selected through at least one pentapeptide KFERQ motif that is recognized by the chaperone HSC70, a member from the Hsp70 protein family (reviewed in [14]). Several proteins associated with neurodegeneration are targeted to CMA, including α-synuclein in Parkinson’s disease [16], amyloid precursor protein in Alzheimer’s disease [17], and TDP43 in ALS [18]. Although there is limited information regarding CMA in ALS pathogenesis, a recent study reported decreased levels of HSC70 in blood cells from sALS patients, associated with increased TDP43 insolubility [19]. In the case of microautophagy, substrates are invaginated by the endosomal/lysosome vesicles in a non-selective or selective way. The selectivity in microautophagy substrates degradation is given by the binding to the same HSC70 chaperone (reviewed in [20]). However, there is no information in the literature regarding microautophagy in ALS.

Macroautophagy, hereinafter simply called autophagy, is an evolutionarily highly conserved catabolic pathway for the degradation of long-lived proteins and whole organelles, which is constitutively active but can also be regulated if necessary [21]. At the beginning of autophagy, a new double-membrane organelle is formed, called autophagosome, which engulfs intracellular cargo. The autophagosomes can fuse directly with lysosomes to generate autolysosomes or first fuse with endosomes to form an intermediate called amphisome, which then fuses with lysosomes [22]. The initiation of autophagy can be activated by the inhibition of mTOR or the activation of AMP-kinase (AMPK), which triggers the formation of the ULK1 complex, composed of ULK1/2, ATG13, FIP200, ATG101 [22]. The ULK1 complex promotes the nucleation of an isolation membrane or phagophore. Phosphorylation of ULK1 promotes phosphoinositol-3-kinase class III (PI3KC3)/Beclin1 complex formation through phosphorylation of Beclin1 [23] and ATG14L [24], which then recruits more phosphoinositol-3-phosphates (PI3Ps) through binding of FYVE domain-containing PI3P-binding proteins to expand the autophagosome membrane. Atg8-homologs are incorporated and support the expansion and closure of the autophagosome as well as the fusion with the lysosome [25,26]. An E3-ligase-like acting complex composed of ATG5, ATG12, and ATG16L mediates the lipidation of Atg8-homologues (LC3 and GABARAP subfamilies) and their binding to the autophagosome membrane [27]. The amount of lipidated Atg8-homologues correlates with the number of autophagosomes formed, hence they are used as indicators of autophagic activity since they are part of autophagosomes. A complex composed of PI3KC3/Beclin1 and UVRAG participates in autophagosome maturation. The action of the PI3KC3/Beclin1/UVRAG complex can be either inhibited by Rubicon or enhanced by Pacer, which antagonize each other [28,29,30,31,32].

In the past, autophagy has been viewed mainly as a non-selective process that engulfs cytosolic content for breakdown to reutilize amino acids and nutrients for cell survival. However, it is now established that autophagy uses defined receptors to recognize, recruit, and target specific cargoes, such as aggregated proteins, damaged mitochondria, or invading pathogens, for degradation at the lysosome to maintain cellular homeostasis [33,34]. These autophagy receptors are characterized by their ability to recognize signals for the degradation on their cargoes, mainly ubiquitin (Ub) [35], as well as simultaneously bind Atg8-homologues on the forming autophagosome or other components from the autophagy machinery, such as the FIP200 (Atg11 orthologue) [36]. E3 ubiquitin ligases attach Ub molecules to a lysine (Lys) on the target protein. Selective autophagy appears to be dependent on the structure and biochemical properties of the autophagy receptor recognizing the cargo since well-characterized autophagy receptors, such as p62/SQSTM1 (p62), optineurin (OPTN), NDP52 (nuclear dot protein 52kDa), and NBR1 (neighbor of BRCA1), contain Ub-binding domains and LC3-interacting regions (LIRs) [37]. However, a type of selective autophagy independent of Ub or LC3 also has been reported [38,39,40]. Post-translational modifications, such as phosphorylation, ubiquitination, and oligomerization, were shown to regulate autophagy receptor binding to the specific substrate/cargo as well as to the autophagy machinery [41]. Autophagy receptor activity is also modulated by autophagy adaptors, which also can bind to Atg8-homologues and can serve as scaffolds for the autophagy machinery [42,43]. Adaptors are Atg8-interacting proteins that participate in different steps of the autophagy pathway, but, unlike receptors, do not bind to ubiquitinated cargos [42]. 

Genetic variants causing functional defects of autophagy receptors and adaptors are increasingly found to be linked to neurodegenerative diseases, including ALS [33]. Impaired or altered functions of autophagy receptors and adaptors in selective autophagy thus may be directly related to the pathogenesis of ALS, which however may also make them potential therapeutic targets for ALS and other neurodegenerative disorders. 

## 2. Autophagy and ALS 

Neurons, like most cell types, display constitutive autophagy activity, which maintains cellular homeostasis [44,45]. Motor neurons are a highly specialized type of neurons and are the cells thought to be most affected during ALS pathology [46]. With their long axons projecting to the muscles, they are the most elongated and polarized mammalian cells. Besides their extreme morphology, motor neurons present a low ability to regenerate and null mitotic activity, being highly reliant on the efficient removal of dysfunctional cellular components such as misfolded proteins or damaged mitochondria to ensure cellular homeostasis and prevent death [47,48]. Neuronal autophagy has a protective function in steady-state conditions, and its loss in mice deficient for genes essential for autophagy, *Atg5*, and *Atg7*, in the central nervous system (CNS) causes neurodegeneration characterized by the widespread accumulation of ubiquitinated cytoplasmic inclusions [49,50]. 

One of the most obvious signs that autophagy is involved in the pathophysiology of ALS is the accumulation of autophagosomes in the cytoplasm of spinal cord neurons of ALS patients [51]. Most insights into the pathophysiologic mechanisms of ALS to date have been obtained by the study of animal models of the disease. Among them, the mutant SOD1 mouse models have been highly studied [32,52,53,54,55]. Murine mutant SOD1 models have shown that there is an increase in the formation of autophagosomes through the visualization of Atg8-homologs, mainly LC3II [56,57], in addition to a decrease in mTOR activity [58]. Simultaneously with the increase in autophagosomes in the SOD1G93A mouse, an increase of p62 levels is usually observed [59,60]. An increase in the transcriptional factor TFEB (regulator of the expression of autophagic and lysosomal biogenesis genes) and Beclin1 is also observed [61]. Collectively, these parameters indicate the activation of the autophagic machinery. However, this activity is not reflected in the degradation of autophagy-cargos in ALS, including protein aggregates and dysfunctional mitochondria [60], probably due to a progressive failure in the lysosomal activity during ALS disease progression [62]. 

A characteristic hallmark of several neurodegenerative diseases, including ALS, is the cytoplasmic translocation and formation of aggregates of various RNA-binding proteins such as TDP43, FUS, hnRNPA1 (heterogeneous nuclear ribonucleoprotein A1), hnRNPA2B1 (heterogeneous nuclear ribonucleoproteins A2/B1), and MATR3, which fulfill essential functions for RNA metabolism [63]. These proteins bind to mRNA by sequestering them, forming membrane-free organelles called ribonucleoprotein granules (RNP), which can be of two types, stress granules (SGs) or P bodies; the formation of these complexes is promoted by low complexity domains present in these proteins [64,65,66]. Mutations in these domains in RNA binding proteins have been related to ALS and favor the formation of aggregates [67]. It has been shown that the formation of these aggregates participates in the pathogenesis of ALS and that the crucial mechanism for the elimination of them is the autophagy process. By inhibiting autophagy through ALS-related mutations, the assembly and disassembly of SGs are altered, thus persisting in diseased cells and acting as initiation points for the formation of toxic protein aggregates [68]. 

Several studies have reported that TDP43 is an independent mTOR autophagy regulator and that, in turn, the activation of autophagy is key for the processing and mitigation of effects caused by the toxic products of TDP43, even in scenarios where this protein lacks mutations [69]. Interestingly, it has been shown that the increase in cytoplasmic TDP43 competes with the 14-3-3 protein and aids the FOXO (Forkhead box) transcriptional factor to translocate to the nucleus, where it promotes protein homeostasis [70]. Furthermore, ALS-linked mutations in the *FUS* gene were shown to deregulate autophagy [71]. The expression of human FUS mutant variants in neuronal cells resulted in the inhibition of autophagosome formation, in addition to the accumulation of ubiquitinated proteins, p62, and NBR1 [71]. A recent study reported that the expression of human wild-type FUS in the murine CNS triggers toxicity associated with RNA metabolism and the inhibition of the autophagy process [72]. The RNA-binding protein hnRNPA1, also plays a role in autophagy and ALS [73]. hnRNPA1 binds to the 3’UTR region of the *Beclin1* mRNA and regulates its expression [73]. In addition, hnRNPA1 actively participates in SG dynamics, a phenomenon related to the occurrence of pathological aggregates in ALS, whose clearing is also mediated by autophagy [74]. 

One of the most common genetic factors in the ALS is the abnormal hexanucleotide expansion in the intronic region of the *C9ORF72* gene [75], which has been related to the alteration of autophagy during ALS pathology on several levels (reviewed in [76]). Recently, the C9ORF72 protein was reported to function in the autophagy pathway [77,78,79,80]; hence, ALS-linked mutations in the *C9ORF72* gene result in decreased C9ORF72 expression and cause alterations in the signaling of autophagic regulators [81]. The reduction of C9ORF72 levels produces, among other effects, a decrease in mTOR activity, increasing the levels of the TFEB transcriptional factor and its translocation from the lysosome to the nucleus, thus resulting in increased lysosomal biogenesis and a higher autophagic flux [82]. Furthermore, the hexanucleotide expansions in the *C9ORF72* gene lead to the formation of nuclear foci of RNA, which can sequester proteins such as TDP43, compromising its function, resulting in disruption of the autophagy–lysosomal pathway [83,84]. 

On the other hand, the role of autophagy seems to be determined by the stage of motor neuron degeneration. In a study exploring the role of autophagy in ALS affected motor neurons, the lack of *Atg7* specifically in motor neurons in a mutant SOD1 model showed unexpected opposing effects at early and advanced stages of the disease [52]. While at the beginning of the neurodegeneration process, autophagy is essential to maintain the neuromuscular junctions, at late stages, autophagy was found deleterious, and the *Atg7* knockout reduced the progression of the disease and increased the life of the mice [52]. These results are further supported by studies that use starvation to increase autophagy at different stages [85]. 

## 3. Selective Autophagy in ALS 

In selective autophagy, selectivity is given by proteins known as autophagic receptors, which are responsible for selecting the substrate and starting the autophagic process (Figure 1). Important cargos targeted to selective autophagy include dysfunctional or damaged mitochondria, protein aggregates, fragments of ER, ribosomes, SGs, microorganisms, among others. Autophagy receptors contain a ubiquitin-binding domain or a zinc finger domain, which facilitates cargo selectivity by binding ubiquitinated proteins [86]. Generally, autophagy receptors also contain an ATG8/LC3-interacting domain, designated as an LC3-interacting region (LIR) motif, which facilitates the attachment of the autophagy receptor to the growing phagophore. Recent studies have shown the presence of a FIP200-interacting region (FIR) in some selective autophagy receptors, allowing the connection of the ubiquitinated cargo with the ULK1/FIP200 autophagy complex, followed by autophagosome formation [36]. Receptors already shown to contain this region are p62 [36], NDP52 [87], and cell-cycle progression gene 1 (CCPG1), a non-canonical receptor [88]. Interestingly, in the p62 receptor, the FIR domain was mapped to contain the LIR domain, suggesting a competition between LC3 and FiP200 to bind it [36].

In ALS, important cargos to be targeted to selective autophagy include protein aggregates, dysfunctional mitochondria, and SGs. Here, we provide an overview of different selective autophagy receptors in which mutations have been associated with ALS development. We also included TBK1, an upstream kinase that phosphorylates several selective autophagy receptors, enhancing the binding between the receptor and the autophagy cargo. Understanding the altered functions of autophagy receptors during ALS pathology may provide insights into their role in the progression of the disease as well as their potential as therapeutic targets for ALS and other neurodegenerative disorders. 

### 3.1. P62 

P62, also known as SQSTM1, was the first autophagic receptor discovered and is capable of binding to ubiquitinated cargos that form aggregates, delivering it to autophagosomes for subsequent degradation, thus maintaining protein homeostasis [89,90]. The level of p62 is usually used as an indicator of autophagic activity [91] since it is degraded together with its cargo by the autophagosome. p62-positive inclusions also have been identified in different variants of ALS [59,92,93,94]. Also, p62 delivers ubiquitinated proteins to the 26s proteasome machinery through an interaction mediated by its ubiquitin-like PB1 domain [95]. Notably, p62 is also abundantly expressed in spinal cord motor neurons [52]. In post-mortem tissue studies, p62 has been shown to be part of ubiquitin-positive inclusions in motor neurons in ALS patients, and as part of neuronal and glial inclusions in FTD patients [59,92,93,94]. Moreover, p62 was reported to directly perform ubiquitin-independent autophagic degradation of mutant SOD1, which recognizes the SOD1 mutant interaction region (SMIR) rather than the UBA domain [13,96]. The recognition of the biological function of p62 and its involvement in neurodegeneration led to a subsequent screening of different cohorts of ALS and FTD patients for p62 mutations [97,98]. Mutations in the p62 gene have been reported to represent approximately 1% of ALS cases [97]. Some of these mutations are in the LIR domain (region of interaction with LC3) of p62, suggesting the contribution of altered selective autophagy to ALS pathogenesis [99]. The high incidence of p62 inclusions in ALS patients suggests that ALS is a misfolded protein disease in which aggregation and phosphorylation of p62 represent a stress response that promotes selective autophagy [100,101,102]. Autophagy-relevant interactions of p62 also involve the N-terminal PB1 domain-mediated oligomerization and/or bind directly to the autophagy receptor NBR1 that selectively targets polyubiquitylated protein aggregates to autophagosomes for degradation [103], and UBA domain-mediated interactions with ubiquitinated OPTN, in order to assemble an autophagy complex that mediates the degradation of ubiquitinated misfolded proteins [104]. The role of NBR1 itself in ALS, however, is not yet explored. Interestingly, the accumulation of p62 and NBR1 together with mutant FUS aggregates in neuronal cells has been shown [71]. Furthermore, post-translational modifications of p62 have been reported to enhance the affinity of the receptor to the cargo [105]. For instance, phosphorylation processes of UBA-domain by the kinase TBK1, which also is involved in ALS (please, see below). Moreover, ubiquitylation of its UBA-domain was shown to disrupt p62 dimerization, facilitating the recognition of polyubiquitinated cargos [106]. 

p62 is also a scaffold protein that acts as a critical regulator in diverse signaling pathways, including amino acid sensing, oxidative stress, DNA damage response, activation of NF-*k*B, cell death and degradation of ubiquitinated proteins via proteasome and autophagy [13,107,108]. In addition to this, recent reports showed that SGs, which are aberrant complexes of RNA and proteins that produce toxicity in the cell and a signature of ALS, are removed specifically via p62-dependent selective autophagy [79,109]. Moreover, *p62* loss of function in zebrafish caused an abnormal locomotor phenotype; it was improved by an mTOR inhibitor (Rapamycin, an autophagy activator) and could also be rescued by wild-type human p62 expression but not by expression of the ALS-linked mutation p62P394L [110]. Furthermore, it was shown that p62 mutations impair selective autophagy and produce neuronal toxicity by interrupting NFE2L.2 (nuclear factor, erythroid derived 2, like 2) function, a protein that regulates oxidative stress and is related to ALS [111,112]. The antioxidative stress response can produce some ALS features in vivo and in vitro [102]. Various p62 mutations are implicated in heterogeneous disease phenotypes, which include ALS, FTD and Paget’s disease of bone [97,98,113]. Mutations in *P62* might affect its function on a more global level, interfering with its normal homeostatic role in removing aggregated proteins, SGs, and cause oxidative stress via autophagy, the failure of which leads to neuronal dysfunction and neurodegeneration [114]. 

One of the mechanisms for activating autophagy due to oxidative stress is the Keap1–Nrf2 pathway, in which, under stress conditions, Nrf2 dissociates from its repressor Keap1, thus allowing the nuclear translocation of Nrf2, where it is able to regulate the expression of several autophagy genes such as p62 [115,116]. Several reports have consolidated this route since alterations of the Keap1–Nrf2 pathway have been found in tissues of animal models of ALS, as well as in postmortem tissues of ALS patients [92].

### 3.2. Optineurin (OPTN)

Optineurin (OPTN) is a ubiquitin-binding protein, containing a UBA domain that is able to simultaneously interact with ubiquitin and LC3, being thus characterized as an autophagy receptor that participates in the removal of mitochondria (mitophagy) [117]. *OPTN* gene mutations were first identified in a small number of Japanese patients with sALS, and post-mortem tissue inclusions of the protein were reported in TDP43 and ubiquitin-positive cytoplasmic inclusions in sALS patients and SOD1-positive inclusions in cases with fALS [102,118]. Strikingly, one of the mutations observed in those patients is a heterozygous E478G missense mutation that impacts the UBAN domain [102,118]. Similar to p62 mutations and the expression of SOD1 mutant protein in zebrafish, depletion of wild-type *OPTN* results in motor dysfunction [39]. Whether or not a dysfunction of *OPTN* is linked with ALS mutations and is due to protein loss of function is under active research. Indeed, recently it was reported that the *OPTN* mutation E478G promotes inflammation and induces neuronal cell death in vitro and in mouse models, which suggests that anti-inflammatory treatment could represent a powerful tool to intervene in the disease progression in ALS patients who harbor mutations in the *OPTN* gene [119]. Additionally, OPTN actively represses the activity of receptor-interacting serine/threonine-protein kinase 1 (RIPK1), a protein that participates in inflammation, apoptosis, and necroptosis. Loss-of-function of OPTN leads to progressive dysmyelination and axonal degeneration through the engagement of necroptotic machinery in the central nervous system (CNS), including RIPK1, RIPK3, and mixed lineage kinase domain-like protein (MLKL) [120], suggesting a crucial role of OPTN in the sensitization to necroptotic death in ALS pathology. 

OPTN was also reported to selectively recognize protein aggregates in an ubiquitin-independent manner through its C-terminal coiled-coil domain [39]. Moreover, this coiled-coil domain, which identifies misfolded proteins in an ubiquitin-independent manner, does not interfere with the capability of OPTN to recognize fALS mutant SOD1 inclusions [39]. Furthermore, a recent study has shown that OPTN, together with NDP52, can be directly recruited to dysfunctional mitochondria by LC3/GABARAP proteins through their LIR domains, independent of ubiquitin-binding [117]. OPTN is found to be the main selective autophagy receptor for ubiquitin-driven mitophagy [117]. OPTN was also shown to promote autophagosomes formation by binding to the Atg12–Atg5–Atg16L complex [121]. On the other hand, OPTN is reported to interact with p62 (see above), accelerating the autophagy flux in cells [122]. ALS-linked mutations in OPTN and OPTN-interacting proteins, such as p62, suggest a common pathogenic pathway. 

### 3.3. Ubiquilin-2 (UBQLN2)

Ubiquilin-2 (UBQLN2), also called *Chap1/Dsk2* or *PLIC*, is a protein that is capable of binding ubiquitinated substrates through three different domains, ubiquitin-associated (UBA) domain, a UBD, and the 26s proteasome complex, via its UBL domain, which is located in its N-terminal, thus acting as a shuttle for delivering ubiquitinated cargo to proteasome [123]. UBQLN2 contains four stress-induced protein 1-like domains (STI-1 like) located between the residues 178–247 and 379–462, which are implicated in the interaction with autophagy proteins. UBQLN2 also has an exclusive PXX domain containing 12 tandem repeats implicated in protein interactions [124,125]. Furthermore, UBQLN2 has implications in protein homeostasis acting in the recognition of misfolded proteins that are eliminated by the proteasome [126]. Nevertheless, UBQLN2 also participates in the degradation of polyubiquitinated proteins that are eliminated by autophagosomes [127]. The function of UBQLN2 in autophagy is not well understood. UBQLN2 is a key protein for the bulk turnover of protein aggregates and damaged organelles indirectly interacting with autophagosome membrane protein LC3 through its UBA domain promoting autophagosome formation and lysosomal fusion [128]. Moreover, a study with transgenic rats with UBQLN2P497H mutation showed reduced levels of LC3-II protein [129]; similar results were reported for ATG7 knockout mice, a protein essential for the autophagosome formation [129]. Additionally, in this study, the rats with UBQLN2P497H mutation developed symptoms similar to ALS and the accumulation of ubiquitinated protein aggregates. Thus, these results suggest that the UBQLN2P497H mutant impairs autophagy–lysosomal pathways and can cause ALS features in rats. 

In ALS, UBQLN2 mutations were first identified in five unrelated patients, suffering from ALS/FTD [125]. In this study, five mutations were reported in the PXX domain, and more than 10 mutations have been reported in other domains of the protein in patients with fALS and sALS [130,131,132,133,134]. UBQLN2 was also found to be colocalized with TDP43 in the spinal cord of sALS patients, making it a component of the motor neuron inclusions in patients affected with the disease, suggesting that UBQLN2 may play an important role in ALS pathology. The observation of ubiquitin, TDP43, and/or FUS in inclusions present in spinal motor neurons with UBQLN2 mutations supports the notion that the build-up of misfolded proteins typically found in ALS-FTD neurons is a downstream consequence of impaired protein degradation pathways [135]. Since UBQLN2 and TDP43 interact and UBQLN2 gain-of-function results in a significant diminution of TDP43 levels in neurons, it was proposed that UBQLN2 functions by improving the clearance and controlling the levels of TDP43 [136]. Remarkably, UBQLN2 also participates in the disease with other ALS-related (and autophagy-related) proteins like VCP (valosin-containing protein) in the process of protein degradation associated with endoplasmic reticulum ERAD [137] and OPTN to modulate the assembly of specific endosomal vesicles [138]. In the case of SGs that participate in ALS, rather than being eliminated with the aggregates protein degradation system via its UBA and UBL domains, UBQLN2 associates with SGs components through its STI1-like region [139]. UBQLN2 influences the early processes of molecular complex dynamics in phase separation that drives SGs formation, suggesting that UBQLN2 regulates the state of the components to be recruited to SGs [139]. UBQLN2 seems a promising target for new therapeutic approaches; more studies are needed to determine the potential impact of the modulation of UBQLN2 levels with therapies in both sALS and fALS. 

### 3.4. Valosin-Containing Protein (VCP)

Valosin-containing protein (VCP) could not be considered an autophagy receptor due to no reported direct interactions with LC3 proteins in mammals. Nevertheless, in yeast, Cdc48 (VCP in mammals) is known to interact indirectly with Atg8 [140]. Furthermore, through its N-domain VCP is able to directly interact with ubiquitin substrates such as protein aggregates since its unfoldase activity is dependent on ATP [141]. The VCP protein is an ATPase implicated in ubiquitin-dependent cellular processes that rely on protein degradation through the UPS or autophagy process [142]. The depletion of VCP leads to autophagosome accumulation rich in ubiquitin, with a failure of autophagosomal maturation following induction of the pathway [143,144]. Mutations affecting the *VCP* gene were previously found to be causative of the Paget disease of the bone and FTD [145]. The effect of VCP mutations on ALS is still under investigation but seems to be related to defects in the RNA-binding proteins TDP43 and FUS, and the accumulation of these inclusions in neurons [146,147]. Moreover, VCP is involved in SG and processing body (P-body) clearance through the autophagy pathway in a process called granulopathy and ALS mutations in this gene impair the process [67]. SGs and P-bodies are cytoplasmic granules that have mRNPs (messenger ribonucleoproteins), complexes consisting of mRNAs and several different proteins that inhibit protein translation [142]. The finding that inhibition of VCP, autophagy, and lysosomes all affect SG clearance highlight an interesting link between VCP mutations, proteostasis, and ALS. 

### 3.5. TANK Binding Kinase 1 (TBK1) 

TANK binding kinase 1 (TBK1) is a serine/threonine kinase described as a participant in several physiological processes, including cellular innate immune response [148], inflammation, cell death mechanisms [149], and selective autophagy [150,151,152]. In 2015, in an extensive collaborative work, researchers sequenced almost 2900 ALS patients and identified *TBK1* as a new gene associated with the disease [153]. *TBK1* mutations are associated with familial and sporadic cases of ALS [154], accounting for almost 1% of the analyzed cases [153]. *TBK1* mutations in ALS occur along the complete coding sequence, with the majority of the cases being missense mutations. Of note, all mutations in *TBK1* found in ALS patients are heterozygotes, supporting the idea that even a mild loss of TBK1 function can contribute to disease pathogenesis [154,155]. These mutations can directly decrease the kinase activity, to reduce protein stability and the interaction with protein partners, including selective receptors for autophagy [156]. TBK1 mediates post-translational modifications in selective autophagy receptors, including OPTN and p62, in their ubiquitin binding-domain and LIR-domains, enhancing the receptor affinity for the cargo or the autophagosome, respectively. In the case of p62, TBK1 is one of the kinases that phosphorylates its UBA cargo-binding domain at Serine-403 [105], strongly enhancing the affinity of the UBA domain of p62 for Lys-48 and Lys-63-associated ubiquitin chains, facilitating the autophagic clearance of p62 and poly-ubiquitinated protein aggregates [102,105]. TBK1 also phosphorylates OPTN at its LIR domain, specifically at Serine-177, increasing the affinity of OPTN to LC3/GABARAP proteins almost 10-folds [150,157]. In mitophagy dependent of ubiquitin, TBK1 is indeed recruited together with OPTN to the depolarized mitochondria, favoring its quick engulfing by the crescent autophagosome [157,158,159]. In ALS-linked mutations in *OPTN* or *TBK1*, a decrease in mitophagy is observed, in which the sequential impairment of TBK1 and -OPTN could contribute to a mitochondria-mediated increase in motor neuron cytotoxicity [159]. TBK1 was also shown recently to phosphorylate directly LC3/GABARAP proteins, participating in the autophagosome formation itself [160]. Thus, *TBK1* mutations could influence some of the key mechanisms that are failing in ALS pathogenesis, including neuroinflammation and proteostasis [161], contributing to the complexity of the disease pathogenesis. In vivo, the injection of AAV-expressing TBK1 in a mutant transgenic mouse model of ALS (SOD1G93A) reduced protein aggregation and extended the lifespan [162], suggesting a potential therapeutic use for TBK1, even in patients lacking *TBK1* mutations. 

## 4. Convergence of Different Pathways in ALS on Autophagy 

Amyotrophic lateral sclerosis (ALS), as a progressive neurodegenerative disease, is associated among other things with a gradual decline of the protein quality control system, including autophagy. Multiple studies suggest that there is a convergence between the associated ALS mechanisms and autophagy [12,163]. One of the pathological features of ALS is the loss of protein homeostasis. Sufficient evidence supports that mutations in the ALS genes *SOD1*, *TDP43,* and *FUS,* among others, result in the incorrect three-dimensional structure of the respective proteins, causing the formation of pathological aggregates containing these proteins [164,165]. In order to maintain cellular homeostasis, autophagy is activated to remove these aggregates [166,167]. On the other hand, it has been reported that protein aggregates and mutations in different genes associated with ALS lead to the assembly of SGs, which also depend on autophagy as a cleaning mechanism to maintain their normal dynamics [67,168]. 

A strong connection between the mechanisms of ALS and autophagy, independent of the presence of protein aggregates, is the regulation of the expression of autophagic genes, produced by RBP associated with ALS [169]. It has been reported that under stress conditions and loss of function, TDP43 can regulate the nuclear translocation of FOXO and TFEB, respectively, in order to promote the transcription of autophagic genes [70]. In turn, TDP43 also has a role in the stability of the mRNAs of genes that participate in the autophagy pathway, as demonstrated in the reduction of the mRNA of *ATG7*, *Raptor*, and *dynactin1*, due to the decrease in TDP43 by RNAi [169,170]. 

Furthermore, it has been described that mutations in *C9ORF72* are related to neurodegeneration associated with ALS and it is suggested that autophagy plays an important role in this process because autophagy is altered in patients who exhibit C9ORF72 repeats, showing accumulated C9ORF72 foci positive for p62 in neuronal cells [171,172]. Studies in cell culture indicate that C9ORF72 regulates autophagy by activating the ULK1 complex and translocation of ULK1 to phagophore, in addition to its interaction with WDR41 and SMCR8 (proteins keys for activation of autophagy by C9ORF72) [173]. Moreover, C9ORF72 interacts with RAB1, RAB5, RAB7, and RAB11 (all proteins involved in autophagy) [77,173,174]. For C9ORF72 to activate the autophagic mechanism, it is required to form a stable complex with WDR41 and SMCR8, thus recruiting the ULK complex and the RABs. Other studies have indicated that the absence of C9ORF72 is also capable of inducing autophagy by deactivating mTOR, increasing the nuclear translocation of TFEB, and thus promoting the transcription of autophagic genes and lysosomal biogenesis [82]. It has been suggested that C9ORF72 performs a function in lysosomes because they colocalize under starvation conditions and, in the absence of C9ORF72, aberrant lysosomes are reported to locate near the nucleus [175]. In summary, the C9ORF72 pathway has a strong connection with the regulation of autophagic activity, but its multifaceted function suggests that new studies are needed to clarify its role. 

Since strong connections between ALS pathways and autophagy are indicated by the above-presented evidence, the points of convergence of these mechanisms may be used as a biological basis for the development of different therapeutic approaches. 

## 5. Therapeutic Approaches Based on Autophagy 

Patients with ALS experience progressive paralysis due to loss of motor neurons. Currently, there are no biomarkers for early diagnosis of the disease, so the diagnosis is carried out in a series of clinical exams after the onset of symptoms, after which most patients do not survive more than five years after diagnosis. The United States Food and Drug Administration (FDA) approved in 1995 the first drug to treat ALS, Riluzole (Rilutek) [176], which decreases excitotoxicity by blocking glutamatergic neurotransmission, non-competitively inhibiting N-methyl-D-aspartate (NMDA) receptors. Edaravone (Radicava) [177], a second drug, was approved 22 years later (in 2017), which acts to eliminate free radicals, reducing oxidative stress. Unfortunately, both drugs are extremely costly and only delay the disease a few months, so the search for early markers of the disease and new chemotherapeutic targets remains an imperative necessity [178]. 

Proteostasis disruption plays an essential role in the development of ALS, and the presence of protein aggregates is a hallmark of the disease; hence, drugs that prevent the accumulation of protein aggregates have recently been developed. A particular target of this strategy is the SOD1 protein. An antisense therapy known as Tofersen (BIIB067) [179], which is a DNA fragment designed to specifically bind to *SOD1* mRNA and trigger its degradation, is currently in phase 3 of clinical trials (NCT02623699 and NCT03070119) [180]. 

Deficiencies in the targeting of neurodegeneration-related proteins to autophagy can be caused by changes in the intrinsic properties of these pathogenic proteins [181]. Factors such as the compactness of the aggregate can affect the efficiency of the assembly of autophagosome proteins. In other cases, the primary problem is at the level of the cargo recognition machinery. Strategies to enhance selective autophagy of pathogenic proteins are still scarce. However, in vitro and in vivo studies have shown that overexpression of the specific autophagy receptor NDP52 significantly reduces the concentration of toxic proteins (i.e., hyperphosphorylated tau) [182]. A similar result was observed under the overexpression of p62 in a tau transgenic mouse [183]. However, in ALS, the overexpression of the p62 receptor in a transgenic mouse model resulted in a worsening in disease parameters, with an earlier onset of the disease, increased protein accumulation, and a shorter life span [54]. 

The current development of science offers new alternatives for the treatment of neurodegenerative diseases such as ALS. The use of antisense oligonucleotides (ASOs) for RNA interference and adeno-associated viral vectors (AAVs) represent attractive therapeutic strategies. In murine models of ALS, with mutations of SOD1, it has been shown that ASO therapy significantly slows the progression of the disease. Efficient gene transduction using AAVs, in particular, serotype 9 (AAV9), of astrocytes and motor neurons have been demonstrated, with promising results in mice and rats for mutant SOD1 [184], where the AAV mediated knockdown of *SOD1* expression delayed the onset of the disease and increased survival of animals [185]. However, mutations in the *SOD1* gene account only for 2% to 5% of ALS cases; the most frequently mutated gene is *C9ORF72*. Using AAV-5-miC, researchers managed to reduce the foci of nuclear RNA that generate toxicity in a murine model of mutant ALS for C9orf72 [186]. Thus, gene therapy appears a promising avenue to slow ALS progression. However, the early diagnosis of ALS remains a scientific and medical challenge. 

Another approach that involves transcriptional modulation directly involves autophagy through the transcription factor HLH-30/TFEB, where it has been shown that by silencing the nuclear export protein XPO-1/XPO1 in a fly ALS model, proteostasis was improved and autophagy and longevity were increased, which was due to a decrease in the nuclear export of HLH-30, thus preventing neurodegeneration. Currently, researchers are investigating the possible use of selective inhibitors of nuclear export (SINE) in ALS, which were previously developed to inhibit XPO1 as a cancer treatment, where good tolerance was found in clinical trials [187]. 

## 6. Conclusions

Despite more than a century and a half of research registered in ALS, and more than 200 clinical trials, no cures or efficient treatments have been developed so far. ALS is a complex, multifactorial disease with genetic and environmental risk factors, with many genes involved in several pathophysiological pathways, which represents a challenge in research. The accumulation of protein aggregates, SGs, and RNA inclusions are features of motor neurons neurodegenerating in ALS. Manipulating the levels of autophagy activity have been explored as therapeutic approaches for treating the disease. However, several of the genes affected in ALS encode proteins related to the autophagy machinery itself, highlighting the relevance of the pathway to motor neuronal homeostasis, but nevertheless limiting interventions based on autophagy as a therapeutic approach. As suggested by recent studies, increasing general autophagy activity could result in a worsening in motor neuron pathogenesis. An interesting strategy could be to increase the activity of specific autophagy receptors, promoting the degradation of specific substrates at a defined timepoint during pathogenesis, possibly in combination with other therapies that can address multiple factors that determine neurodegeneration in patients with ALS via different approaches such as stem cell therapies or gene therapies.

## Figures and Tables

**Figure 1 cells-09-00381-f001:**
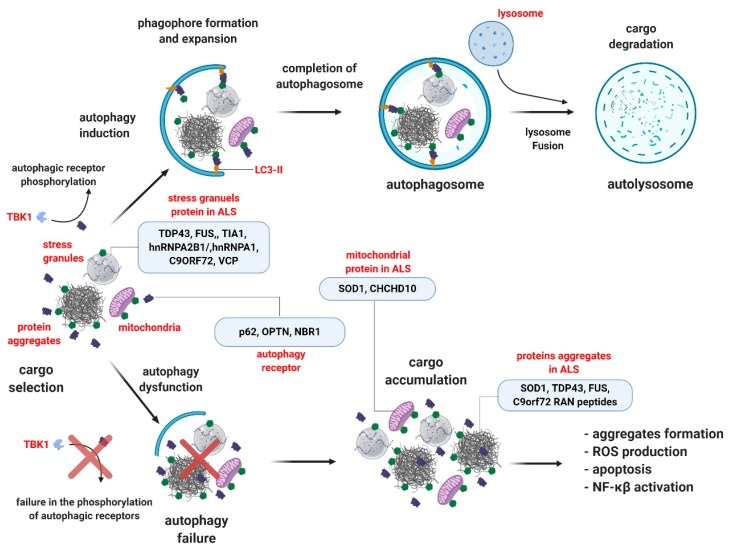
Selective autophagy under physiological and ALS pathological conditions. Protein aggregates, stress granules, and dysfunctional mitochondria are substrates for selective autophagy degradation. Under physiological conditions (upper panel), substrates are bound by selective autophagy receptors, such as p62, OPTN and NBR1 (represented in dark blue) via ubiquitin-binding domains (ubiquitin, in green). Selective autophagy receptors associate with LC3 proteins in the autophagosome (represented in yellow), or other members from the autophagy machinery. Posttranslational modifications in the receptors can enhance the binding with ubiquitinated substrates or the LC3 protein. TBK1 is one of the main kinases acting in this process. The cargo-receptor-LC3 complexes are then sequestered by de novo double-membrane vesicles called the autophagosome, which fuses with the lysosome for the final degradation. Under ALS conditions (lower panel), the failure in selective autophagy can occur through mutations in the genes encoding the receptors themselves or in the TBK1 gene, reducing the activity of the pathway, promoting the accumulation of toxic substrates for motor neurons (Image created with BioRender.com).

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
