# Peer review of "Implications of Selective Autophagy Dysfunction for ALS Pathology"

_cells, 2020, doi:10.3390/cells9020381_

Round 1

Reviewer 1 Report

In this review article, the authors are providing a comprehension of the disease amyotrophic lateral sclerosis (ALS) and autophagy dysfunction. Here they are describing how selective autophagy deregulation and mutations associated with this involved in the ALS disease. The review is well written and has covered the subject in the most completeness. However, the following concerns need to be addressed.

(1)    Line 51, Nuclear protein transport also gets affected in the ALS which causes cytosolic accumulation of the proteins, as an important mechanism of initiation of the ALS. Also, the accumulation of the proteins and not having enough machinery to degrade all the accumulated proteins could be one mechanism even though autophagy remains functioning with regular way, unless also possible where stress granules and proteins translation/related mRNAs involved in the autophagy process itself get halted in stress granules and undergo solid fibrils formation. This information should be included with most recent references.

(2)    chaperone-mediated autophagy (CMA) and microautophagy, should the author provide any information on how or why these two forms of autophagy are different and not involved in the ALS?

(3)    Line 124, The neurons are not regenerating whether they are elongated or not so possibly the damage of neurons and not replaceable or difficult to replace is more imminent reason to have the functioning is major issue, irrespective of the size and shape, that’s the reason in ALS first there is loss of the memory and cognitive activity and eventually muscle functioning.

Minor issue:

(4)    Line 96 “……. lysosome to maintain cellular homeostasis.” This sentence should have references.

(5)    Line 35 ……………. associated with it (Ghasemi and Brown 2017). The reference should be accompanied by the most recent reviews and articles.

(6)    Line 66-67 formatting issue. Broken continuity.

(7)    Line 223 Formatting issue, 281, 315, 363, 384 are abbreviations, full names wherever available could be provided. Is that line 384 TBK1 as 5. TBK1?

Author Response

Reviewer 1

In this review article, the authors are providing a comprehension of the disease amyotrophic lateral sclerosis (ALS) and autophagy dysfunction. Here they are describing how selective autophagy deregulation and mutations associated with this involved in the ALS disease. The review is well written and has covered the subject in the most completeness. However, the following concerns need to be addressed.

(1)    Line 51, Nuclear protein transport also gets affected in the ALS which causes cytosolic accumulation of the proteins, as an important mechanism of initiation of the ALS. Also, the accumulation of the proteins and not having enough machinery to degrade all the accumulated proteins could be one mechanism even though autophagy remains functioning with regular way, unless also possible where stress granules and proteins translation/related mRNAs involved in the autophagy process itself get halted in stress granules and undergo solid fibrils formation. This information should be included with most recent references.

Response: We thank the reviewer for her/his comments and time. We have modified this sentence to include the suggestions of the reviewer and have added a recent review that captures those suggestions.

Mechanisms found impaired in ALS mainly involve four pathways, RNA metabolism and translation, protein quality control and degradation, mitochondrial function, and transport and trafficking via the cytoskeleton within and between compartments (Taylor, Brown, and Cleveland 2016, Burk and Pasterkamp 2019, Gautam et al. 2019)(Cook and Petrucelli 2019).

Cook, C. and L. Petrucelli (2019). "Genetic Convergence Brings Clarity to the Enigmatic Red Line in ALS." Neuron 101(6): 1057-1069.

(2)    chaperone-mediated autophagy (CMA) and microautophagy, should the author provide any information on how or why these two forms of autophagy are different and not involved in the ALS?

Response: We thank the reviewer for her/his suggestion, we added the following section at line 68 in the manuscript including the corresponding references.

CMA cargos are selected through at least one pentapeptide KFERQ motif that is recognized by the chaperone HSC70, a member from the Hsp70 protein family (reviewed in (Kaushik and Cuervo 2018)). Several proteins associated with neurodegeneration are targeted to CMA, including α-synuclein in Parkinson's disease (Cuervo, Stefanis et al. 2004), amyloid precursor protein in Alzheimer’s disease (Park, Kim et al. 2016), and TDP43 in ALS (Huang, Bose et al. 2014). Although there is limited information regarding CMA in ALS pathogenesis, a recent study reported decreased levels of HSC70 in blood cells from sALS patients, associated with increased TDP43 insolubility (Arosio, Cristofani et al. 2019). In the case of microautophagy, substrates are invaginated by the endosomal/lysosome vesicles, in a non-selective or selective way. The selectivity in microautophagy substrates degradation is given by the binding to the same HSC70 chaperone (reviewed in (Tekirdag and Cuervo 2018)). However, there is no information in the literature regarding microautophagy in ALS.

Arosio, A., R. Cristofani, O. Pansarasa, V. Crippa, C. Riva, R. Sirtori, V. Rodriguez-Menendez, N. Riva, F. Gerardi, C. Lunetta, C. Cereda, A. Poletti, C. Ferrarese, L. Tremolizzo and G. Sala (2019). "HSC70 expression is reduced in lymphomonocytes of sporadic ALS patients and contributes to TDP-43 accumulation." Amyotroph Lateral Scler Frontotemporal Degener: 1-12.

Cuervo, A. M., L. Stefanis, R. Fredenburg, P. T. Lansbury and D. Sulzer (2004). "Impaired degradation of mutant alpha-synuclein by chaperone-mediated autophagy." Science 305(5688): 1292-1295.

Huang, C. C., J. K. Bose, P. Majumder, K. H. Lee, J. T. Huang, J. K. Huang and C. K. Shen (2014). "Metabolism and mis-metabolism of the neuropathological signature protein TDP-43." J Cell Sci 127(Pt 14): 3024-3038.

Kaushik, S. and A. M. Cuervo (2018). "The coming of age of chaperone-mediated autophagy." Nat Rev Mol Cell Biol 19(6): 365-381.

Park, J. S., D. H. Kim and S. Y. Yoon (2016). "Regulation of amyloid precursor protein processing by its KFERQ motif." BMB Rep 49(6): 337-342.

Tekirdag, K. and A. M. Cuervo (2018). "Chaperone-mediated autophagy and endosomal microautophagy: Joint by a chaperone." J Biol Chem 293(15): 5414-5424.

(3)    Line 124, The neurons are not regenerating whether they are elongated or not so possibly the damage of neurons and not replaceable or difficult to replace is more imminent reason to have the functioning is major issue, irrespective of the size and shape, that’s the reason in ALS first there is loss of the memory and cognitive activity and eventually muscle functioning.

Response:  We thank the reviewer for advising us to modify this sentence at line 138 in the revised manuscript.

With their long axons projecting to the muscles, they are the most elongated and polarized mammalian cells. Besides their extreme morphology, motor neurons present a low ability to regenerate and null mitotic activity, being highly reliant on the efficient removal of dysfunctional cellular components such as misfolded proteins or damaged mitochondria to ensure cellular homeostasis and prevent death (Ciechanover and Kwon 2015, Kanning, Kaplan, and Henderson 2010).

Minor issue:

(4)    Line 96 “……. lysosome to maintain cellular homeostasis.” This sentence should have references.

Response: We have added 2 references.

However, it is now established that autophagy uses defined receptors to recognize, recruit and target specific cargoes, such as aggregated proteins, damaged mitochondria, or invading pathogens, for degradation at the lysosome to maintain cellular homeostasis (Deng, Purtell et al. 2017, Johansen and Lamark 2020).

Deng, Z., K. Purtell, V. Lachance, M. S. Wold, S. Chen and Z. Yue (2017). "Autophagy Receptors and Neurodegenerative Diseases." Trends Cell Biol 27(7): 491-504.

Johansen, T. and T. Lamark (2020). "Selective Autophagy: ATG8 Family Proteins, LIR Motifs and Cargo Receptors." J Mol Biol 432(1): 80-103.

(5)    Line 35 ……………. associated with it (Ghasemi and Brown 2017). The reference should be accompanied by the most recent reviews and articles.

Response: We have included a more recent reviews on the genetics of ALS.

Like other neurodegenerative diseases, ALS is a complex disease with a growing number of genes and loci associated with it (Ghasemi and Brown 2017)(Brenner and Weishaupt 2019, Cook and Petrucelli 2019, Mejzini, Flynn et al. 2019).

Brenner, D. and J. H. Weishaupt (2019). "Update on amyotrophic lateral sclerosis genetics." Curr Opin Neurol 32(5): 735-739.

Cook, C. and L. Petrucelli (2019). "Genetic Convergence Brings Clarity to the Enigmatic Red Line in ALS." Neuron 101(6): 1057-1069.

Mejzini, R., L. L. Flynn, I. L. Pitout, S. Fletcher, S. D. Wilton and P. A. Akkari (2019). "ALS Genetics, Mechanisms, and Therapeutics: Where Are We Now?" Front Neurosci 13: 1310.

(6)    Line 66-67 formatting issue. Broken continuity.

Response: We have fixed the issue.

(7)    Line 223 Formatting issue, 281, 315, 363, 384 are abbreviations, full names wherever available could be provided. Is that line 384 TBK1 as 5. TBK1?

Response: We have revised the manuscript to include full names to abbreviations whenever they appear the first time. 

Reviewer 2 Report

The paper entitled Implications of selective autophagy dysfunction for ALS pathology is a very exhaustive review, properly written and overwhelmingly well referenced.

Author Response

We thank the reviewer for her/his comments.

We addressed some english language issues throughout the manuscript.

Reviewer 3 Report

Dear Authors,

This is a great review explaining the role of autophagy in ALS and potential therapeutic implications. 

The review is well-developed, however I missed the “conclusions” section and maybe a figure/table to make it more readable.

Thank you very much for your contribution.

Author Response

Response: We appreciate the comments and suggestions of the reviewer. We had included a figure before however there was a mistake during submission, we apologize to the reviewer. Now the figure is included in the manuscript. We also added a conclusion section as was requested by the reviewer, please see below.

Conclusions

      Despite more than a century and a half of research registered in ALS, and more than 200 clinical trials, no cures or efficient treatments have been developed so far. ALS is a complex, multifactorial disease with genetic and environmental risk factors, with many genes involved in several pathophysiological pathways, which represents a challenge in research. The accumulation of protein aggregates, SGs, and RNA inclusions are features of motor neurons neurodegenerating in ALS. Manipulating the levels of autophagy activity has been explored as therapeutic approaches for treating the disease. However, several of the genes affected in ALS encode proteins related to the autophagy machinery itself, highlighting the relevance of the pathway to motor neuronal homeostasis, nevertheless limiting interventions based on autophagy as a therapeutic approach. As suggested by recent studies, increasing the general autophagy activity could result in a worsening in motor neuron pathogenesis. An interesting strategy could be to increase the activity of specific autophagy receptors, promoting the degradation of specific substrates at a defined timepoint during pathogenesis, possibly in combination with other therapies that can address multiple factors that determine neurodegeneration in patients with ALS via different approaches such as stem cell therapies or gene therapies.

Round 2

Reviewer 1 Report

Article can be accepted for the publication.